# Effects of a Fully Humanized Type II Anti-CD20 Monoclonal Antibody on Peripheral and CNS B Cells in a Transgenic Mouse Model of Multiple Sclerosis

**DOI:** 10.3390/ijms23063172

**Published:** 2022-03-15

**Authors:** Sabine Tacke, Rittika Chunder, Verena Schropp, Eduard Urich, Stefanie Kuerten

**Affiliations:** 1Institute of Anatomy and Cell Biology, Friedrich-Alexander University Erlangen-Nürnberg (FAU), 91054 Erlangen, Germany; sabine.ta@t-online.de (S.T.); rittika.chunder@uni-bonn.de (R.C.); verena.schropp@fau.de (V.S.); 2Roche Pharmaceutical Research and Early Development, Neuroscience, Roche Innovation Center, 4070 Basel, Switzerland; eduard.urich@roche.com; 3Institute of Neuroanatomy, Medical Faculty, University of Bonn, 53115 Bonn, Germany

**Keywords:** B cells, CD20, experimental autoimmune encephalomyelitis, multiple sclerosis, obinutuzumab, rituximab

## Abstract

Successful therapy with anti-CD20 monoclonal antibodies (mAbs) has reinforced the key role of B cells in the immunopathology of multiple sclerosis (MS). This study aimed to determine the effects of a novel class of anti-CD20 mAbs on vascular and extravascular central nervous system (CNS)-infiltrating B cells in experimental autoimmune encephalomyelitis (EAE), an animal model of MS. Male hCD20xhIgR3 mice and wild-type C57BL/6 (B6) mice were immunized with human myelin oligodendrocyte glycoprotein (MOG)_1–125_ to induce EAE. While hCD20xhIgR3 mice were injected intravenously with an anti-human CD20 mAb (5 mg/kg) (rituximab (a type I anti-CD20 mAb) or obinutuzumab (a type II anti-CD20 mAb), B6 mice received the anti-mouse CD20 antibody 18B12. Neither mAb affected clinical disease or serum antibody levels. Obinutuzumab and rituximab had an impact on splenic and CNS-infiltrated B cells with slightly differential depletion efficacy. Additionally, obinutuzumab had beneficial effects on spinal cord myelination. B cell depletion rates in the 18B12/B6 model were comparable with those observed in obinutuzumab-treated hCD20xhIgR3 mice. Our results demonstrate the usefulness of anti-CD20 mAbs for the modulation of B cell-driven peripheral immune response and CNS pathology, with type II antibodies potentially being superior to type I in the depletion of tissue-infiltrating B cells.

## 1. Introduction

Multiple sclerosis (MS) is a chronic inflammatory demyelinating disease of the central nervous system (CNS) and the most common cause of irreversible neurologic disability in young adults [1]. MS is considered to be a heterogeneous disease, with complex genetics and multiple environmental factors playing a role in individual susceptibility to disease development [2,3]. The clinical course of MS is dependent on the subtype of the disease, with the classical variants including: (i) relapsing-remitting MS (RRMS), which accounts for 80–90% of all MS cases; (ii) secondary progressive MS (SPMS), which follows 70–80% of RRMS cases 10–15 years after disease onset; and (iii) primary progressive MS, which is the rarer variant accounting for 10–15% of all cases [4,5]. Also in the progressive forms of MS occasional relapse episodes of intensified symptoms may occur [6].

Although the detrimental role of inflammation in MS is undisputed, the pathogenic cell populations and the precise roles they play remain controversial [7,8]. Furthermore, another level of complexity is added by the diverse mechanisms driving the development of the disease, not just among the different subtypes but also among patients within one type of MS [9,10]. It is generally agreed that the inflammatory reaction in MS results from the cumulative effect of a number of factors, including activities of cells of the innate and adaptive immune system and their mediators, and effector molecules such as cytokines and antibodies [11,12].

One of the best-characterized animal models capable of reflecting important aspects of MS, among which are both autoimmune inflammatory and neurodegenerative processes, is experimental autoimmune encephalomyelitis (EAE) [13,14]. EAE can be induced in susceptible animal strains by active immunization with components of the myelin sheath or by the passive transfer of autoreactive T cells [15]. Traditional EAE models are mostly dependent on T cells and macrophages; in particular, T-helper cells have been the focus of research. This “T cell paradigm” was subsequently also assumed for human MS. Nevertheless, despite being dubbed a “T cell-mediated disease”, B cells have increasingly gained importance and attention as key players in the pathogenesis of MS [16,17].

The updated conceptual understanding of the involvement of B cells in the immunopathophysiology of MS has mainly emerged on the basis of the success of anti-CD20 therapy in treating patients with RRMS [18]. In initial studies, depletion of circulating B cells by rituximab, a chimeric anti-CD20 monoclonal antibody (mAb), led to a rapid reduction in gadolinium-enhancing lesions and magnetic resonance imaging lesion load, as well as a decrease in relapse activity in patients with RRMS [10]. More recent studies involving humanized (ocrelizumab) and human (ofatumumab) anti-CD20 mAbs confirmed a high level of efficacy in RRMS in the clinic [19,20,21,22]. Furthermore, in patients with primary progressive MS, ocrelizumab is associated with lower rates of clinical and magnetic resonance imaging progression compared with placebo [23]. Despite these therapeutic developments, SPMS—in which one of the characteristic features is chronic CNS-compartmentalized inflammation—remains difficult to treat [24,25].

One of the factors contributing to sustained neuroinflammation within the CNS has been identified as the presence of large B cell aggregates that are present in the inflamed meninges of a substantial portion of patients with SPMS [26,27,28,29,30]. These meningeal B cell-rich aggregates of immune cells are considered as potential mediators of “sequestered” inflammation that spreads the continuous CNS injury underlying the secondary progressive form of the disease [17,31].

Clinical studies have demonstrated that B cells residing in these specific CNS compartments are protected from depletion following conventional anti-CD20 therapy [32,33]. Intrathecal injection of rituximab in patients with low-inflammatory SPMS resulted in incomplete depletion of B cells within the cerebrospinal fluid compartment and insufficient dampening of neuroinflammation [32]. Certain B cell-depleting characteristics of the anti-CD20 mAb (rituximab) used in these studies could be one of the reasons for the partial resistance of B cells found within the CNS of these patients. Briefly, depending on their B cell-depleting characteristics, anti-CD20 mAbs are grouped into type I and type II [34]. While type I mAbs (including rituximab, ocrelizumab, and ofatumumab) activate the classical pathway of the complement cascade, and induce complement-dependent cytotoxicity by inducing the reorganization of CD20 into lipid rafts, type II mAbs trigger direct cell death upon binding CD20 without cross-linking by secondary antibodies [34,35,36]. Antibody-dependent cell-mediated cytotoxicity and antibody-dependent phagocytosis can be induced by both antibody types in the presence of immune effector cells. However, type II mAbs exert more potent natural killer cell-mediated antibody-dependent cell-mediated cytotoxicity and increased monocyte- and macrophage-mediated antibody-dependent phagocytosis compared with type I mAbs, particularly when glycoengineered [37,38,39].

Obinutuzumab is a humanized type II anti-CD20 mAb that is currently approved as a first-line treatment for chronic lymphocytic leukemia and rituximab-refractory follicular lymphoma [40,41,42]. Compared with rituximab, obinutuzumab has demonstrated an increased depletion rate of peripheral B cells in healthy donors, as well as a more efficient reduction in malignant B cells in patients with chronic lymphocytic leukemia [40,43]. The elbow hinge modification of obinutuzumab results in an alternative binding conformation of the CD20–obinutuzumab complex and is responsible for the more potent induction of direct cell death compared with rituximab and ofatumumab [35,44]. Furthermore, complement-dependent cytotoxicity and, hence, FcγRIIb-mediated CD20 internalization are reduced compared with rituximab, resulting in sustained availability and activity of obinutuzumab [39,45].

In the current study, we tested the therapeutic efficacy of the type II mAb obinutuzumab compared with that of the type I mAb rituximab in a B cell-dependent EAE model using a double transgenic (dbtg) mouse line that expresses both mouse and human CD20 on B cells and tolerates the administration and presence of human immunoglobulin (Ig)G [46,47]. A particular focus of the study was directed toward the effects of the therapeutic monoclonal antibodies on B cell infiltrates in the CNS, which is a feature of chronic EAE [48].

## 2. Results

### 2.1. No Significant Change in Clinical Disease following Anti-CD20 mAb Treatment in Chronic EAE

C57BL/6 (B6) and hCD20xhIgR3 mice (*n* = 5–6 mice per group) were immunized with human myelin oligodendrocyte glycoprotein (MOG)_1–125_ to induce both T cell- and B cell-dependent EAE [49]. Clinical EAE was scored daily. In all mice, immunization caused severe disease with a chronic course that was characterized by complete paralysis of the hind limbs (Figure 1). Details on disease onset and severity in the different groups are provided in Table 1. To investigate the effect of anti-CD20 mAb treatment in chronic EAE, mice were treated either with obinutuzumab (hCD20xhIgR3), rituximab (hCD20xhIgR3), 18B12 (wild-type [WT] B6), or the matching isotype controls, 19 days after they had reached an EAE score of ≥2.5. Treatment was repeated on days 22 and 25. None of the anti-CD20 mAbs were able to improve clinical disease as determined by EAE score (Figure 1, Table 1) or total body weight (Table 1).

### 2.2. No Impact of Anti-CD20 mAb Treatment on Total and Anti-MOG_1–125_ Serum IgG

To evaluate whether the depletion of CD20^+^ cells by obinutuzumab, rituximab, or the mouse mAb 18B12 had an effect on serum Ig levels, total IgG and anti-MOG_1–125_ IgG were measured by enzyme-linked immunosorbent assay (ELISA). Blood was taken from *n* = 5–6 mice per group, 7 days after the end of treatment. As shown in Figure 2, neither total IgG nor anti-MOG_1–125_ antibody titers were diminished following treatment with anti-CD20 mAbs, compared with the corresponding isotype controls.

### 2.3. Targeting of B Cell Subsets in the Spleen by Anti-CD20 mAb Treatment

Flow cytometric analysis of the spleen was performed to investigate the depleting capacity of the different anti-CD20 mAbs. Splenic B cells were isolated from 5 to 6 mice per group, 7 days after the end of treatment with anti-CD20 mAbs or corresponding isotype controls. As shown in Figure 3, obinutuzumab-treated mice displayed decreased numbers of naïve (CD19^+^/IgM^−/^IgD^+^) (Figure 3B), marginal zone (CD19^+^/CD93^−^/CD21^hi^/CD1d^hi^) (Figure 3C), follicular (CD19^+^/CD93^−^/CD21^int^/CD1d^lo^) (Figure 3D), and isotype-switched (CD19^+^/IgG^+^) B cells (Figure 3G). Rituximab treatment mainly depleted isotype-switched B cells (Figure 3G). 18B12, which was used as mouse anti-CD20 mAb in WT B6 mice, was highly potent and depleted approximately 90% of all B cell subsets (Figure 3A; *p* ≤ 0.001 vs. isotype control (Welch’s *t*-test)) except for memory B cells and marginal zone B cells (Figure 3B–D; naïve B cells: *p* ≤ 0.01 vs. isotype control (Mann–Whitney *U* test); follicular B cells: *p* ≤ 0.01 vs. isotype control (Welch’s *t*-test)). Interestingly, although memory (CD19^+^/CD80^+^/CD73^+^) B cells express CD20, they were spared by anti-CD20 mAb-mediated depletion in all three treatment groups (Figure 3F). The relative depletion compared to the isotype control is provided in Figure 3I for each group.

### 2.4. Reduction in the Number of Perivascular and Parenchymal CNS B Cell Infiltrates in Anti-CD20 mAb-Treated Mice

Two types of infiltrates were analyzed: diffuse parenchymal vs. dense perivascular B cell infiltrates in the cerebellum (Figure 4A). As above, different groups of mice were treated with obinutuzumab, rituximab, 18B12, or the matched isotype controls and sacrificed 7 days after the end of treatment. Compared with treatment with the isotype controls, the numbers of both diffuse and dense B cell infiltrates were reduced following treatment with obinutuzumab and 18B12 (Figure 4B and Figure 5A,C) although statistical significance was not reached. In contrast, rituximab only depleted dense perivascular B cell infiltrates but did not target B cells infiltrating into the parenchyma (Figure 5A,C). Obinutuzumab treatment also reduced the number of diffuse T cell infiltrates, while none of the anti-CD20 mAbs had any effect on dense T cell infiltrates (Figure 5B,C).

### 2.5. Positive Effect of Obinutuzumab Treatment on Spinal Cord Myelination

Transmission electron microscopy analysis of mouse spinal cord sections revealed irreversible and ongoing axonal degeneration and abnormal myelination in all animals (Figure 6). Analysis was performed during the chronic stage of the disease, 7 days after the last treatment with anti-CD20 mAbs or the corresponding isotype controls. Neither obinutuzumab nor rituximab treatment had an effect on axons that were in the process of degeneration or had already degenerated (Figure 6B, left and middle panels). However, there was a reduction in the number of axons displaying signs of abnormal myelination, such as splitting or ballooning of myelin lamellae, in mice treated with rituximab (*p* = 0.0185) (Figure 6B, middle panel). As shown in Figure 6C, the total number of axons per mm^2^ (as an indicator of axonal loss) remained unaffected by anti-CD20 treatment, as did the g-ratio (the diameter of the axon divided by the diameter of the nerve fiber including myelin sheath, Figure 6D). Analysis of the overall degree of myelination demonstrated a significant shift back toward normally myelinated axons in obinutuzumab-treated mice (Figure 6E, left panel; *p* = 0.0208 for the slope difference between the control and treatment group). Representative images of EM pathology in the different groups are shown in Figure 7.

## 3. Discussion

Emerging evidence suggests that B cells contribute to MS pathogenesis in multiple ways [50] that encompass antibody production, pro- and anti-inflammatory cytokine secretion, as well as antigen presentation [51]. Studies performed in different EAE models have helped to further understand the mechanisms underlying this B cell involvement. For instance, the antigen presenting capacity of B cells has been highlighted in rMOG protein-induced EAE where activated B cells aided in the differentiation and proliferation of T_H_1 and T_H_17 cells [52]. In an adoptive transfer model of EAE, it has been demonstrated that the development of an autoimmune attack on the CNS was preceded by the induction of major histocompatibility complex (MHC) II expression on B cells [53]. Studies conducted in B cell-specific MHC II knockout mice confirmed that B cells provide critical cellular functions necessary for the development of EAE [54]. Finally, another study highlighted that non-selective B cell depletion using anti-CD20 therapy eliminated preexisting B regulatory cells, which were crucial for limiting disease progression [55]. In recent years, treatment with anti-CD20 mAbs has demonstrated compelling evidence of a profound reduction in new brain lesions and relapse rates in patients with RRMS. All of the anti-CD20 mAbs studied have been of type I classification. While these mAbs deplete peripheral B cells efficiently, they do not sufficiently target CD20^+^ B cells that reside within the CNS compartments [32,33].

In this study, we examined the effects of the type I anti-CD20 mAb rituximab and the type II anti-CD20 mAb obinutuzumab on vascular and extravascular CNS-infiltrated B cells in the B cell-dependent human MOG protein-induced EAE model.

Additionally, the aim of our study was to evaluate the impact of obinutuzumab compared to other anti-CD20 antibodies on established B cell infiltration within the CNS. We have previously demonstrated that B cell infiltration in B cell-dependent EAE mainly occurs in the cerebellum around 20–30 days after disease induction [56,57]. Accordingly, we used a therapeutic treatment protocol and focused on the later EAE stages for all subsequent analyses.

Numerous studies using anti-CD20 antibodies and other drugs [52,58,59,60] have shown that early treatment (before EAE onset or at the time point of EAE onset) typically prevents or significantly diminishes EAE development onset. This is not surprising because early interference with the immune response induced by immunization also affects the later development of CNS pathology. In our study no improvement of clinical EAE was observed after anti-CD20 antibody treatment due to the fact that treatment was initiated after the peak of disease. Indeed, axonal damage occurs early on in the EAE model and is the major reason for irreversible clinical deficits. Unless a treatment is administered that is neuroregenerative, no improvement in clinical score can be expected. In further support of our study, Brand et al., 2021 [61] also did not observe any changes in the clinical course of spontaneous chronic EAE after anti-CD20 treatment despite efficient depletion of B cells in the periphery.

These and our data could be particularly important for understanding the long-term effects of B cell depletion therapy in MS patients, which are still unknown given that the first anti-CD20 antibody for MS treatment was only approved recently.

Analysis of splenic B cell subsets revealed more potent depletion of CD19^+^ B cells by obinutuzumab compared with rituximab, possibly due to the differentiated effector functions of type II mAbs [35,39,62]. There are controversial reports regarding the depletion of B cells within secondary lymphoid organs in humans. In a study on patients with systemic lupus erythematodes and autoimmune thrombocytopenia, a complete depletion of splenic B cells was observed [63]. In another study involving renal transplant surgery patients, a single dose of rituximab did not achieve complete depletion of B cells in secondary lymphoid organs but altered their phenotype and function [64]. The lack of efficient memory B cell depletion in the spleen in our study can be confirmed by Häusler et al., 2018, where the authors have demonstrated that memory B cells can escape systemic anti-CD20-mediated B cell depletion in peripheral organs [55]. Furthermore, our data show that follicular B cells were more efficiently depleted than marginal zone B cells, which is in line with previous findings [65]. Since marginal zone B cells play an important role in the first-line defense against systemic blood-borne infections [66], their preservation under anti-CD20 therapy should be beneficial. Marginal zone B cells may be partially spared from depletion because both obinutuzumab and rituximab are IgG1 antibodies; it was previously shown that this isotype depletes marginal zone B cells inefficiently compared with IgG2a, which depletes 98% of follicular and marginal zone B cells [65]. In addition, the expression of CD20 per cell varies between the different B cell subsets dependent on their maturation and differentiation stage as well as activation status [67], which potentially alters their relative susceptibility to antibody-mediated depletion. Plasma cells do not express CD20 and are, hence, not targeted by anti-CD20 mAbs, which is consistent with observations made in this study, both in flow cytometry experiments and when measuring total serum IgG and anti-MOG_1–125_ antibody titers. Since bone marrow plasma cells are intrinsically long-lived, and thus mature B cells are not required for maintaining their numbers, short-term CD20^+^ B cell depletion has no effect on pre-existing antibody levels [68].

A key finding of our study was that both B and T cells that infiltrated into the CNS parenchyma were reduced by obinutuzumab, but spared in rituximab-treated mice. This is consistent with the most recent findings of Roodselaar and colleagues, who demonstrated more pronounced depletion of B and T cells in the brain when using type II anti-CD20 mAbs in a mouse model of SPMS [62]. In patients with MS, B cell depletion is associated with less CNS inflammation over the months following treatment [19,23,69,70,71,72]. B cells not only function as antigen-presenting cells but are also an important source of soluble inflammatory mediators, which can be toxic to neurons and oligodendrocytes [73,74]. Consistent with the more potent depletion of parenchymal B cells, electron microscopic analysis revealed a significantly positive effect of obinutuzumab on the myelination status of the nerve fibers in the inflamed spinal cord.

In summary, our results suggest that type II mAbs such as obinutuzumab might be superior to type I mAbs with respect to exerting effects on extravascular B cells. Our data also show that the mouse anti-CD20 mAb 18B12 is highly effective in reducing the number of CNS-infiltrating B cells, and might thus be a good murine option for studying the depletion of tissue-infiltrating B cells using WT B6 EAE models. In the current study, treatment was started in the stage of established disease. It remains to be determined whether differences between type I and II mAbs are more pronounced when administered shortly before or during acute disease, in which inflammatory processes prevail and neurodegeneration is only at its starting point. Furthermore, we only examined a relatively short therapeutic time frame during which we did not observe any effects of mAb treatment on clinical disease and antibody levels. It is conceivable that longer observation periods will be better suited to reveal potential treatment outcomes.

## 4. Materials and Methods

### 4.1. Mice

Male 6-week-old WT B6 mice were purchased from Charles River Laboratories (Sulzfeld, Germany) and dbtg hCD20xhIgR3 mice were obtained from Taconic (Ejby, Denmark). The dbtg mice expressed both human and mouse CD20 on their B cells, in addition to an Ig mini-repertoire composed of the secreted forms of H-γ1, IgL-κ, and IgL-λ chains [46,47]. Interestingly, in our hands, EAE was inducible only in male hCD20xhIgR3 mice, while female mice were resistant to disease development. Mice were housed under pathogen-free conditions with a 12 h light/dark cycle and free access to standard rodent diet (ssniff Spezialdiäten, Soest, Germany) and water. Special care such as softened food and ClearH_2_O HydroGel (ClearH_2_O, Portland, ME, USA) was provided for paralyzed animals. All animal experiments were approved by the Regierung von Unterfranken (file number 55.2-2532-2-577) and performed according to the criteria outlined by the German Animal Welfare Law. They complied with the German Law on the Protection of Animals, the “Principles of Laboratory Animal Care” (NIH publication no. 86-23, revised 1985), and the ARRIVE guidelines for reporting animal research [75].

### 4.2. Active EAE Induction and Clinical Assessment

All animals were 15 weeks old at the time of immunization. When establishing the model in hCD20xhIgR3 mice, this age proved to be ideal for EAE induction. Mice were immunized subcutaneously into both sides of the flank with a total dose of 200 µg human recombinant MOG_1–125_ emulsified in complete Freund’s adjuvant (induction kit from Hooke Laboratories, Lawrence, MA, USA). Pertussis toxin (100 ng; Hooke Laboratories) diluted in 100 µL of sterile phosphate-buffered saline (PBS) was injected intraperitoneally on day 0 and again 24 h later. Signs of EAE were monitored daily, starting at day 5 after immunization, using the standard EAE scoring system with the following grades: 0, no signs of disease; 1, limp tail; 2, hindlimb weakness; 3, nearly complete or complete hindlimb paralysis; 4, complete hindlimb and partial forelimb paralysis; 5, moribund. Increments of 0.5 were used to account for intermediate scores not clearly demarcated by the five categories.

### 4.3. Anti-CD20 mAb Treatment

The anti-CD20 IgG1 mAbs obinutuzumab and rituximab, the isotype-matched control for obinutuzumab (hIgG1), and the mouse anti-CD20 IgG2a antibody 18B12 were kindly provided by F. Hoffmann-La Roche (Basel, Switzerland). The chimeric IgG1 control for rituximab (chIgG1) was purchased from Absolute Antibody (Oxford, UK) and the isotype control antibody for 18B12 (muIgG2a) from Bio X Cell (Lebanon, NH, USA). Mice were grouped (*n* = 5–6 per group) according to their EAE course and score, and treatment was started 19 days after they had reached an EAE score of ≥2.5. Antibodies were injected intravenously into the lateral tail vein at 5 mg/kg body weight, diluted in 50 µL sterile PBS. Treatment was repeated on days 22 and 25. Animals were euthanized 7 days after the last treatment by administration of CO_2_.

### 4.4. Serum Collection and Antibody ELISA

Blood was taken from the inferior vena cava of each mouse and allowed to clot. Serum was then collected by centrifugation at 1000× *g* for 15 min at 4 °C and stored at −80 °C until further analysis. For quantification of total serum IgG and human MOG_1–125_-specific IgG, ELISAs were performed using the Mouse IgG Total Uncoated ELISA kit (Thermo Fisher Scientific, Waltham, MA, USA) and the SensoLyte anti-human MOG (1–125) mouse IgG-specific quantitative ELISA kit (AnaSpec, Fremont, CA, USA) following the manufacturers’ instructions.

### 4.5. Isolation of Splenocytes for Flow Cytometry

Mouse spleens were dissected and cells isolated by filtering the tissue homogenate through a 70 µm cell strainer. Cells from individual mice were suspended in 30 mL Gibco RPMI 1640 medium (Merck, Darmstadt, Germany) supplemented with 10% fetal bovine serum (Thermo Fisher Scientific), 50 µM β-mercaptoethanol (Sigma-Aldrich, St. Louis, MO, USA), and 20 mM 4-(2-hydroxyethyl)-1-piperazine ethanesulfonic acid (HEPES; Sigma-Aldrich), and centrifuged at 500× *g* for 5 min. Resuspended cells were treated with 1× red blood cell lysis buffer (BioLegend, London, UK) for 10 min to remove red blood cells. The reaction was stopped by adding PBS and cells were washed twice. Cells were filtered through a 70 µm cell strainer to remove clumps of dead cells. After an additional centrifugation step, cells were resuspended at a concentration of 2.5 × 10^6^/mL for all further procedures.

### 4.6. Flow Cytometry

Splenocytes were harvested as described above and 5 × 10^5^ cells were transferred into a 96-well plate and incubated with 0.2 µL/well of BD Horizon Fixable Viability Stain 780 (FVS780; BD Biosciences, San Jose, CA, USA) for 30 min. Cells were centrifuged at 500× *g* for 5 min before incubation with 50 µL master mix containing 0.125 µg of each antibody. Five different antibody panels were used to identify cell types. Allophycocyanin (APC) anti-CD19 (BioLegend) and brilliant violet (BV) 510 anti-CD3e (BD Biosciences) were used for B cell and T cell staining, respectively. BV510 anti-CD19, APC anti-CD93, BV421 anti-CD21/CD35, and phycoerythrin (PE) anti-CD1d (all BioLegend) were used to differentiate between CD19^+^/CD93^−^/CD21^int^/CD1d^lo^ follicular B cells and CD19^+^/CD93^−^/CD21^hi^/CD1d^hi^ marginal zone B cells. Plasma cells were stained with BV605 anti-CD45R as well as BV421 anti-CD138 (both BD Biosciences) and defined as CD45R^−^/CD138^+^. BV510 anti-CD19, BV421 anti-CD80 (BD Biosciences), and PE anti-CD73 (BD Biosciences) were used to stain CD19^+^/CD80^+^/CD73^+^ memory B cells. BV510 anti-CD19, APC anti-IgM (BioLegend), PE anti-IgD (BioLegend), and BV421 anti-IgG (BioLegend) marked CD19^+^/IgM^−^/IgD^+^ naïve and CD19^+^/IgG^+^ isotype-switched cells. After incubation, stained cells were washed with BD FACSFlow (BD Biosciences) twice and resuspended in a final volume of 150 µL BD FACSFlow.

Flow cytometric acquisition was performed on a CytoFLEX S machine equipped with CytExpert 2.2 software (Beckman Coulter, Brea, CA, USA). The gating strategy is displayed in Figure 8. Doublets were excluded using the combined width parameter of the forward and side scatter. Viable cells were gated before lymphocyte gates were set for further analysis. Data analysis was performed using FlowJo v10.0.6 (FlowJo, LLC, Ashland, OR, USA).

### 4.7. Immunohistochemistry and Histopathology of Cerebellum

To determine the effects of type I and type II anti-CD20 mAbs on the depletion of CNS-infiltrating B cells, the cerebellum was studied during the chronic stage of EAE. In our previous studies, the cerebellum showed the highest degree of B cell infiltration compared with other regions of the CNS [48]. After transcardial perfusion with 4% paraformaldehyde (Carl Roth, Karlsruhe, Germany), mouse cerebella were dissected and post-fixed in the same fixative overnight, followed by washing with phosphate buffer (PB), then dehydration and embedding in paraffin. A total of 150 serial sections were cut at a thickness of 5 µm, and every fifth section (up to a total of 30 sections) was stained for B and T cells. In preparation for staining, sections were deparaffinized with xylene and rehydrated in an ascending series of isopropanol. For antigen retrieval, sections were boiled in 10 mM sodium citrate buffer (pH 6.0) for 15 min. All the following incubation steps were performed at room temperature. After washing with tris-buffered saline (TBS), the cerebellar tissue sections were blocked with 5% milk powder (Heirler, Radolfzell, Germany) in TBS with 0.1% Tween 20 detergent (TBST) for 30 min, followed by 1 h incubation with monoclonal rabbit anti-mouse anti-CD3 antibody (SP1629, 1:150; Abcam, Cambridge, UK). Slides were then washed twice with TBST followed by 1 h incubation with a biotinylated goat anti-rabbit IgG antibody (1:500; Abcam). Alkaline phosphatase coupled to streptavidin (1:500; Vector Laboratories, Burlingame, CA, USA) was applied for 30 min after washing with TBST. CD3^+^ cells were visualized with the Vector BCIP/NBT substrate kit (Vector Laboratories). Before B220^+^ cells were stained on the same tissues, sections were washed for 15 min with TBST. Subsequently, slides were incubated with monoclonal rat anti-mouse anti-CD45R antibody (RA3-6B2, 1:250; Thermo Fisher Scientific) for 1 h, followed by a further 1 h incubation with biotinylated goat anti-rat IgG antibody (1:500; Abcam). Slides were washed three times with TBST between every step. Streptavidin–horseradish peroxidase (1:2000, Abcam) was applied for 35 min. 3,3’-diaminobenzidine (Vector Laboratories) was used as a chromogen and tissues were counterstained with Nuclear Fast Red (Sigma-Aldrich). Blinded analysis was conducted by counting infiltrates per section using a Leica DM2000 light microscope (Leica Microsystems, Wetzlar, Germany). Perivascular cuffing of cells was defined as dense infiltrates; cells that infiltrated loosely into the parenchyma were categorized as diffuse infiltrates. Infiltrates with fewer than five cells were excluded from the analysis.

### 4.8. Transmission Electron Microscopy of Spinal Cord

To assess ultrastructural spinal cord pathology, two segments of the lumbar spinal cord region from each mouse were post-fixed in 2.5% glutaraldehyde (Carl Roth) in 0.1 M sodium cacodylate buffer (pH 7.4; Carl Roth) overnight, followed by washing with PB until further processing. Osmification of tissues was performed with 1% osmium tetroxide (Science Services, München, Germany) and 1.5% potassium ferrocyanide (Merck) for 2 h. After washing with PB overnight, tissues were dehydrated in an ascending ethanol series until 100% ethanol was replaced by 100% acetone, which was eventually replaced by epoxy resin embedding medium. Specimens were polymerized overnight at 60 °C in embedding molds filled with 100% epon (Carl Roth) containing 2% glycidether hardener (Carl Roth). Ultrathin sections (50 nm) were cut and placed on copper mesh grids for transmission electron microscopy analysis. Sections were then stained with uranyl acetate for 10 min and lead citrate for 10 min.

A total of 25 images per mouse of the ventral funiculus surrounding the median fissure were acquired using a Carl Zeiss Leo EM 906 E transmission electron microscope (Zeiss, Oberkochen, Germany) equipped with ImageSP software v1.2.10.2 (Sysprog, Minsk, Belarus and Tröndle Restlichtverstärkersysteme, Moorenweis, Germany). All images were taken at a magnification of 3597×. Quantification of axonal degeneration was performed manually using ImageSP and by categorizing axons into three different groups: (i) abnormally myelinated axons; (ii) degenerating axons; and (iii) already degenerated axons. In addition, the number of myelinated axons per square millimeter (mm^2^) was calculated on the same images. Nine images and a mean of 594 ± 70 randomly selected axons per animal (*n* = 5–6 mice per group) were analyzed. The correlation between myelin sheath thickness and axon diameter was assessed by measuring the axonal and the total nerve fiber diameter maxima, from which the g-ratio was subsequently calculated by dividing the diameter of the axon by the total nerve fiber diameter. A mean of 229 ± 37 randomly selected axons in 6–7 images per mouse (*n* = 5–6 mice per group) were measured. Analysis was performed blinded.

### 4.9. Statistical Analysis

Analysis was performed using GraphPad Prism 9.0.0 (GraphPad, San Diego, CA, USA). The statistical tests used for each analysis are provided in the figure legends. Only significant differences are indicated in the figures. A *p*-value of ≤0.05 was considered statistically significant.

## Figures and Tables

**Figure 1 ijms-23-03172-f001:**
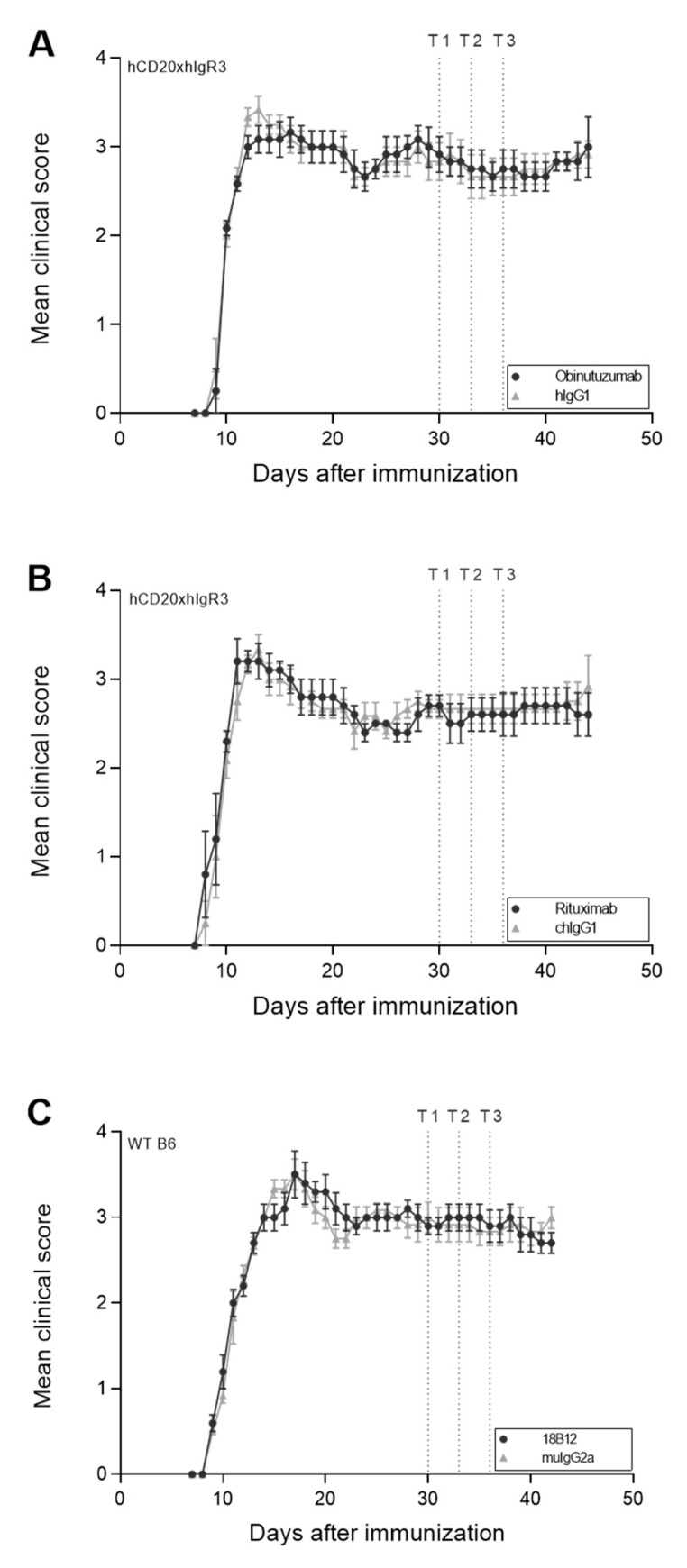
Effect of anti-CD20 mAb treatment on clinical disease in hCD20xhIgR3 and WT B6 mice. EAE was assessed daily in all mice (*n* = 5–6 per group) and scores are shown as means ± standard error of the mean. Dotted lines mark the three time points (T1, day 19; T2, day 22; T3, day 25) of treatment with 5 mg/kg anti-CD20 mAb: (**A**) obinutuzumab, (**B**) rituximab, or (**C**) 18B12, or their respective isotype control antibodies (hIgG1, chIgG1, or muIgG2a). Note that treatment started 19 days after the mice had reached an EAE score of ≥2.5. Mice were scored using the standard EAE scale ranging from 0 to 5: 0, no signs of disease; 1, limp tail; 2, hindlimb weakness; 3, nearly complete or complete hindlimb paralysis; 4, complete hindlimb and partial forelimb paralysis; 5, moribund. The area under the curve (AUC) was calculated for each group. To evaluate whether there was a statistically significant difference in disease severity between the isotype control- and anti-CD20 mAb-treated groups the *p*-value of the AUCs was calculated using an unpaired *t*-test. No statistical significance was observed.

**Figure 2 ijms-23-03172-f002:**
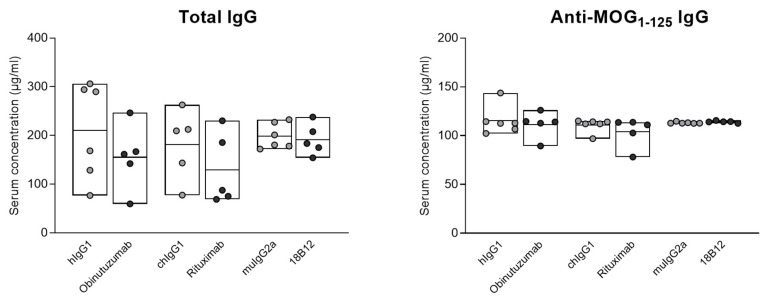
Serum levels of total and anti-MOG_1–125_ IgG in hCD20xhIgR3 and WT B6 mice. Mice were each injected three times with 5 mg/kg anti-CD20 mAbs or the corresponding isotype controls. Serum IgG levels were determined by ELISA in 5–6 mice per group, 7 days after the last injection. Each circle represents the mean value for an individual mouse. The bars indicate the group mean value as well as minimum and maximum values. Statistical significance was evaluated by unpaired *t*-test. There was no statistically significant difference between any of the groups.

**Figure 3 ijms-23-03172-f003:**
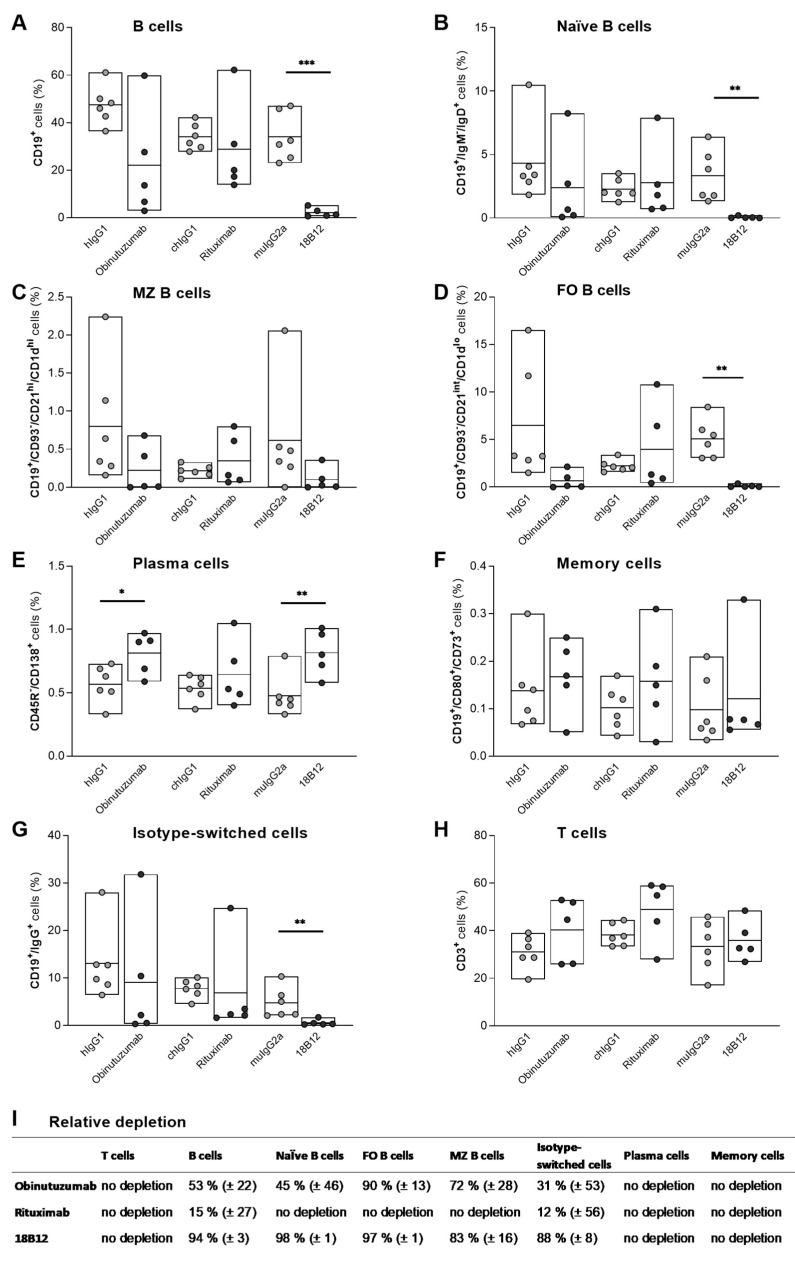
Depletion of splenic B cell subsets by anti-CD20 mAbs in hCD20xhIgR3 and WT B6 mice. Spleen cells were harvested from 5 to 6 animals per group, 7 days after injection with 5 mg/kg of obinutuzumab, rituximab, 18B12, or the corresponding isotype controls. Using flow cytometry, cells were gated on single live lymphocytes followed by the identification of different B cell subsets using five separate antibody panels. Percentages were calculated in reference to the total lymphocyte population. (**A**) B cells were defined as CD19^+^, (**B**) naïve B cells as CD19^+^/IgM^−^/IgD^+^, (**C**) marginal zone B cells as CD19^+^/CD93^−^/CD21^hi^/CD1d^hi^, (**D**) follicular B cells as CD19^+^/CD93^−^/CD21^int^/CD1d^lo^, (**E**) plasma cells as CD45R^−^/CD138^+^, (**F**) memory cells as CD19^+^/CD80^+^/CD73^+^, and (**G**) isotype-switched cells as CD19^+^/IgG^+^. (**H**) T cells were defined as CD3^+^. Each circle represents the mean value of an individual mouse. The bars indicate the group mean value as well as minimum and maximum values. The following statistical tests were used for group comparisons: panels (**A**,**D**), Welch’s *t*-test; panels (**B**,**G**), Mann–Whitney *U* test; panel (**C**), unpaired *t*-test (OBZ), Welch’s *t*-test (RTX) and Mann–Whitney *U* test (18B12); panels (**E**,**F**), unpaired *t*-test; panel (**H**), unpaired *t*-test (OBZ); Welch’s *t*-test (RTX) and unpaired *t*-test (18B12). * *p* ≤ 0.05, ** *p* ≤ 0.01, *** *p* ≤ 0.001. (**I**) Relative depletion (mean ± SEM) of the different B cell subsets in anti-CD20 mAb- compared to isotype control-treated groups. FO, follicular; MZ, marginal zone.

**Figure 4 ijms-23-03172-f004:**
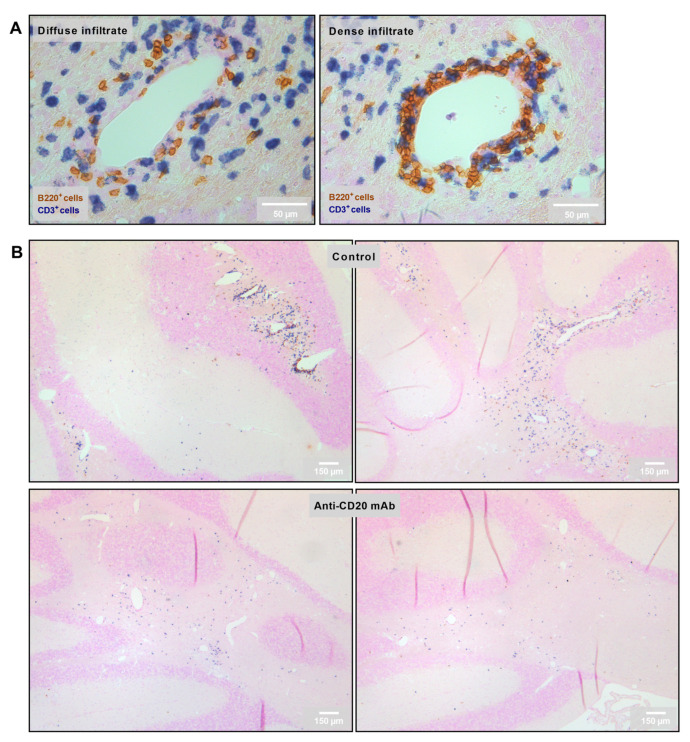
Representative images of CNS infiltration in isotype control- and anti-CD20 mAb-treated groups: (**A**) Categorization of infiltrates depending on the diffuse vs. dense infiltration of lymphocytes into the CNS. B220^+^ cells were defined as B cells, and CD3^+^ cells as T cells. (**B**) Representative images of cerebellar infiltration in isotype control- (upper row) vs. anti-CD20 mAb (lower row)-treated groups.

**Figure 5 ijms-23-03172-f005:**
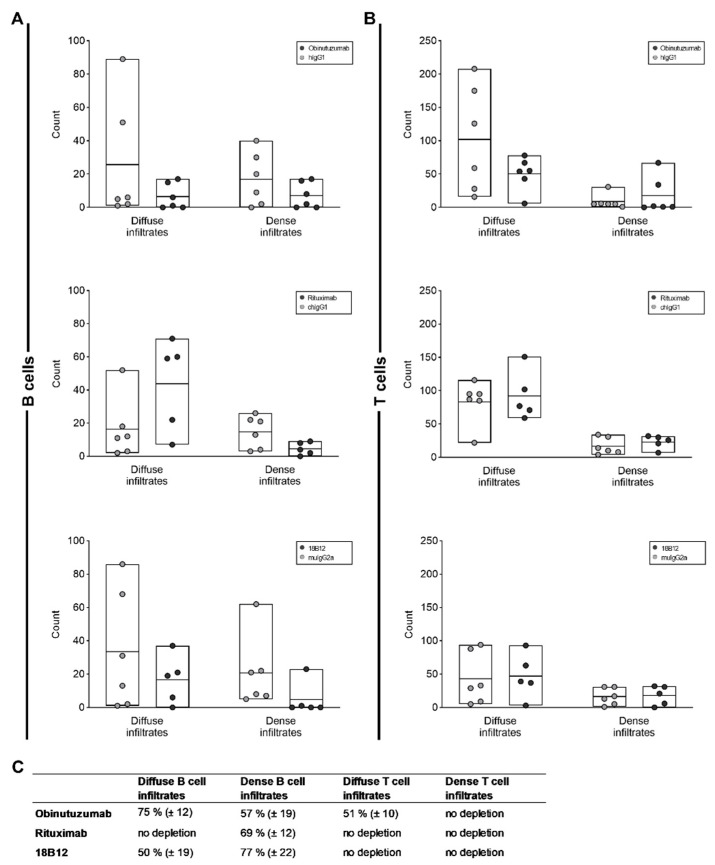
Effect of anti-CD20 mAbs on CNS-infiltrating B and T cells. The cerebella of 5–6 mice per group were analyzed 7 days after the end of treatment with anti-CD20 mAbs or isotype controls, as indicated in the figure insets. Thirty images per mouse were analyzed. Each circle represents the mean value for an individual mouse. The bars indicate the group mean value as well as minimum and maximum values. The following statistical tests were used for group comparisons: diffuse infiltrates in panel (**A**), Mann–Whitney *U* test (OBZ, RTX) and unpaired *t*-test (18B12); dense infiltrates in panel (**A**), unpaired *t*-test (OBZ, RTX) and Mann–Whitney *U* test (18B12); diffuse infiltrates in panel (**B**), unpaired *t*-test; dense infiltrates in panel (**B**), Mann–Whitney *U* test (OBZ) and unpaired *t*-test (RTX, 18B12). There was no statistically significant difference between any of the groups. (**C**) Relative depletion (mean ± SEM) of B cell and T cell infiltrates in anti-CD20 mAb- compared to isotype control-treated groups.

**Figure 6 ijms-23-03172-f006:**
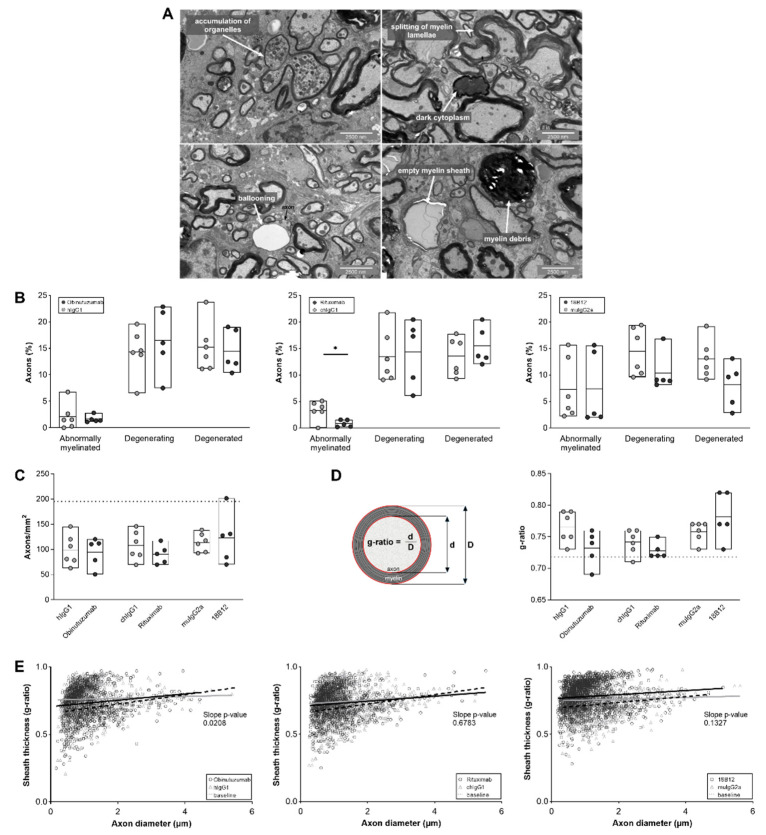
Anti-CD20 mAb treatment effects on ultrastructural spinal cord pathology in hCD20xhIgR3 and WT B6 mice. Spinal cords of mice (*n* = 5–6 per group) were dissected 7 days after the final injection of 5 mg/kg anti-CD20 mAbs or corresponding isotype control. (**A**) Representative electron micrographs showing the different categories of nerve fiber degeneration. Splitting and ballooning of myelin lamellae indicate abnormally myelinated axons; accumulation of organelles and dark cytoplasm indicate degenerating axons; and myelin debris and empty myelin sheaths indicate degenerated axons. (**B**) Quantification of axonal degeneration. Each circle represents the mean value of an individual mouse. The bars indicate the group mean value as well as minimum and maximum values. The following statistical tests were used for group comparisons: abnormal axons, unpaired *t*-test (OBZ) and Welch’s *t*-test (RTX, 18B12); degenerating and degenerated axons, Welch’s *t*-test (all groups). * *p* ≤ 0.05. (**C**) Number of axons per mm^2^. The dotted line marks the baseline value (195 axons/mm^2^) in healthy mice (*n* = 7). Statistical significance was evaluated by unpaired *t*-test. (**D**) The g-ratio was calculated by dividing the diameter of the axon by the diameter of the nerve fiber (axon + myelin sheath) in a mean ± SD of 229 ± 37 randomly selected axons/mouse (*n*
*=* 5–6 mice per group). The dotted line represents the baseline value of normally myelinated axons (0.718) (*n* = 7; mean ± SD of 13 ± 19 axons/mouse). The bars indicate the group mean value as well as minimum and maximum values. Statistical significance was calculated using unpaired *t*-test. (**E**) Scatter plots showing the myelin sheath thickness of individual myelinated axons as a function of the respective axon diameter with simple linear regression analysis (black line: anti-CD20 mAbs; grey line: isotype controls; dashed line: baseline). The baseline was determined from healthy mice (*n* = 7). The significant slope change in the regression analysis of obinutuzumab-treated mice toward the baseline indicates a positive effect of the therapeutic mAb on myelination (*p* = 0.0298). Statistical significance was calculated using ANCOVA.

**Figure 7 ijms-23-03172-f007:**
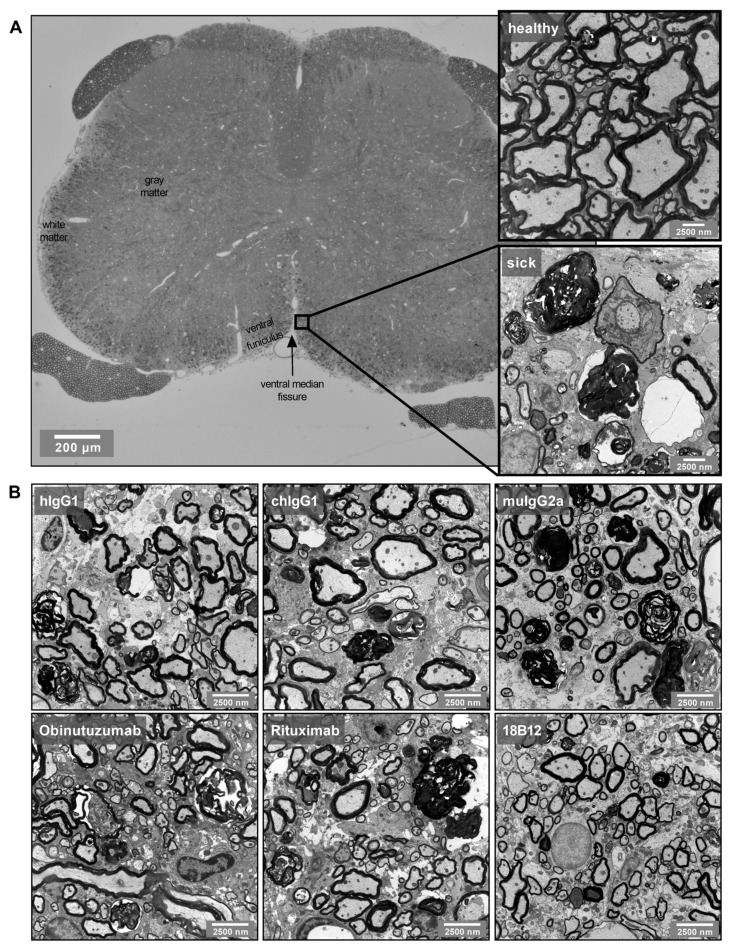
Representative spinal cord EM images: (**A**) EM images were taken from the ventral funiculus of transverse spinal cord sections. The magnifications show the difference between healthy and EAE (sick) tissue. (**B**) Representative EM images from the different isotype control- and anti-CD20 mAb-treated groups.

**Figure 8 ijms-23-03172-f008:**
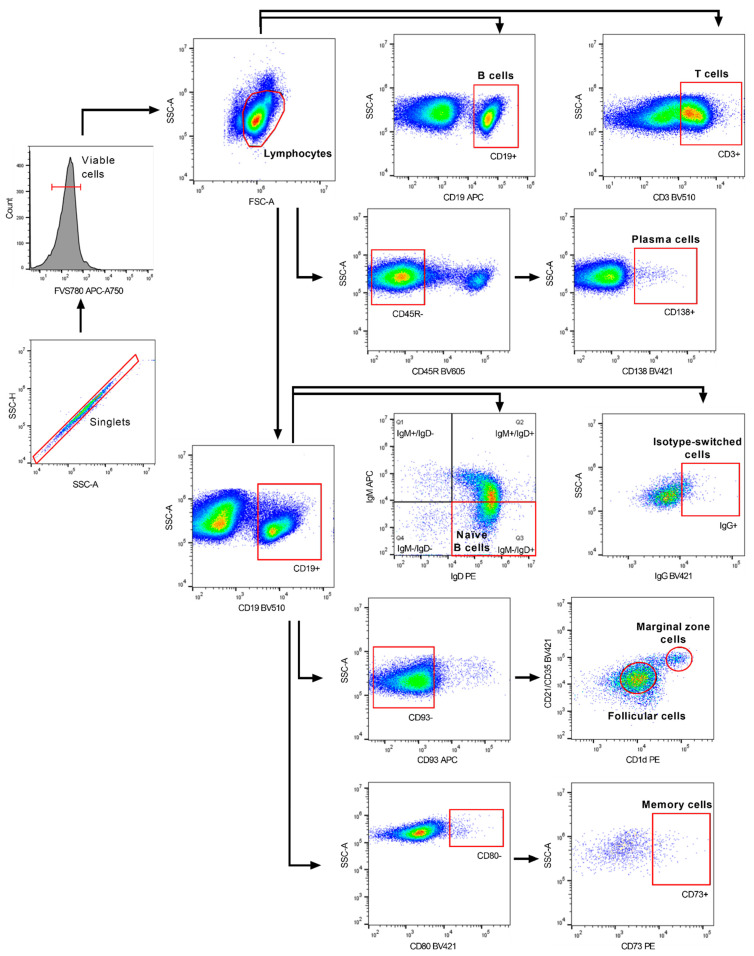
Representative flow cytometric gating strategy for the analysis of B cell subsets. Cells were gated on single live lymphocytes followed by the identification of the different B cell subsets using five separate antibody panels.

**Table 1 ijms-23-03172-t001:** Clinical disease characteristics in mice treated with anti-CD20 mAbs.

Mouse Strain	hCD20xhIgR3		hCD20xhIgR3		WT B6	
Treatment	hIgG1	obinutuzumab	chIgG1	rituximab	muIgG2a	18B12
Number of mice (*n*)	6	6	6	5	6	5
Day of EAE onset	9.7 ± 0.5	9.8 ± 0.4	9.3 ± 0.8	9.0 ± 1.0	9.0 ± 0.0	9.0 ± 0.0
Maximum EAE disease score	3.5 ± 0.3	3.6 ± 0.5	3.6 ± 0.6	3.5 ± 0.4	3.7 ± 0.3	3.8 ± 0.3
Disease score before start of treatment	2.8 ± 0.5	3.0 ± 0.5	2.7 ± 0.3	2.7 ± 0.3	3.0 ± 0.4	2.9 ± 0.2
Final disease score	2.9 ± 0.4	3.0 ± 0.8	2.9 ± 0.9	2.6 ± 0.5	3.0 ± 0.3	2.7 ± 0.3
Score difference ^a^	0.1 ± 0.4	0.0 ± 1.0	0.3 ± 0.9	−0.1 ± 0.4	0.0 ± 0.4	−0.2 ± 0.2
Weight before start of treatment (g)	26.8 ± 1.7	26.1 ± 0.8	26.4 ± 1.8	26.5 ± 2.1	26.0 ± 1.4	26.9 ± 1.7
Final weight (g)	27.0 ± 2.1	25.8 ± 0.8	27.2 ± 1.7	26.7 ± 2.6	25.8 ± 2.0	26.6 ± 2.6
Weight difference ^a^ (g)	0.3 ± 0.8	−0.3 ± 0.4	0.7 ± 0.6	0.2 ± 0.9	−0.2 ± 1.7	−0.3 ± 1.5

^a^ Versus value before start of treatment; Data shown are mean values ± standard deviation; EAE, experimental autoimmune encephalomyelitis; WT, wild-type.

## Data Availability

The datasets used and analyzed during the current study are available from the corresponding author upon reasonable request.

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
