# Peer review of "Effects of a Fully Humanized Type II Anti-CD20 Monoclonal Antibody on Peripheral and CNS B Cells in a Transgenic Mouse Model of Multiple Sclerosis"

_ijms, 2022, doi:10.3390/ijms23063172_

Round 1

Reviewer 1 Report

The authors have addressed my comments appropriately. 

Reviewer 2 Report

This study investigated the therapeutic efficacy of the type II mAb obinutuzumab compared with that of the type I mAb rituximab in a B cell-dependent EAE model using a double transgenic mouse line that expresses both mouse and human CD20 on B cells and tolerates the administration and presence of human IgGs. Neither mAb affected clinical disease or serum antibody levels. Obinutuzumab and rituximab had an impact on splenic and CNS-infiltrated B cells with slightly differential depletion efficacy. Additionally, obinutuzumab had beneficial effects on spinal cord myelination. B-cell depletion rates in the 18B12/B6 model were comparable with those observed in obinutuzumab-treated hCD20xhIgR3 mice. It was demonstrated the usefulness of anti-CD20 mAbs for the modulation of B cell-driven peripheral immune response and CNS pathology, with type II antibodies potentially being superior to type I in the depletion of tissue-infiltrating B cells. This study was organized well, and this paper discussed effects of type II anti-CD20 mAbs in EAE based on the current results.

This manuscript is a resubmission of an earlier submission. The following is a list of the peer review reports and author responses from that submission.

Round 1

Reviewer 1 Report

Timely and very relevant. Would be interesting to see if variable responses in animals exposed to corona virus with obinutuzumab vs rituximab. 

Additional comments: I believe these results do demonstrate the usefulness of anti-CD20 mAbs for the modulation of B cell-driven peripheral immune response and CNS pathology, with type II antibodies potentially being superior to type I in the depletion of tissue-infiltrating B cells.

Author Response

We would like to thank the reviewer for the positive evaluation of our manuscript. Indeed, it would be interesting to study obinutuzumab and rituximab in the context of a coronavirus infection. We are currently conducting a clinical trial in ocrelizumab-treated MS patients, aiming to understand the long-term success of coronavirus vaccination after anti-CD20 therapy. 

Reviewer 2 Report

The study by Tacke et al., focuses on the effects of anti-CD20 monoclonal antibodies on the transgenic mouse model of multiple sclerosis. There are many interesting and new data in this manuscript regarding disease severity, antibodies level, splenic B-cell variations, CNS infiltration and myelin condition following the treatment.

In general; The main finding is that all the monoclonal antibodies used, did not affect the severity of the disease, yet, splenic B-cells populations and neuropathology were influenced. There are major questions that arise regarding the planning of this study and the timing chosen for initiation of treatment in the disease course, starting at one-time point, about 20 days after disease onset, without optimizing the treatment initiation. The time that was selected for animal sacrifice and sampling of cells and tissues must also be calibrated/ optimized.

Thus, the findings regarding changes in cell populations and the neuropathological outcome did not correlate or reflect the disease severity.

Specific comments:

Introduction:

3 different subtypes of MS mentioned, however, there is a 4th type; progressive relapsing (PRMS) that is not mentioned. (see another comment in the discussion).

Results:

Expressions used for demonstrating the results:

For example: Line 114, (beginning of the results) " no improvement…" , it actually describes that  there is no significant change in disease severity.

Line 151 "no depletion…" when there are no significant changes/reductions in splenic B-cell content.

Statistics: p values are missing in the results. In most figures the statistic is difficult to follow; which group is compared with which.

Figure 3, demonstrates how the cell sorting experiments were performed, it is technical and does not provide real data, if at all, this figure may appear in the methods section.

Discussion:

The discussion is very short and not in depth, particularly it lacks explanations of why there is no correlation between the changes that occurred in cell population and myelin content and disease severity.

Lately, the significant role of B-cells in MS and EAE received much attention and many studies have been performed and published investigated this subject, the authors did not elaborate enough on this topic in the discussion nor in the introduction.

Author Response

We thank the reviewer for the positive and detailed evaluation of our manuscript and we are happy to address the remaining concerns in the following.

2.1) There are major questions that arise regarding the planning of this study and the timing chosen for initiation of treatment in the disease course, starting at one-time point, about 20 days after disease onset, without optimizing the treatment initiation. The time that was selected for animal sacrifice and sampling of cells and tissues must also be calibrated/ optimized. Thus, the findings regarding changes in cell populations and the neuropathological outcome did not correlate or reflect the disease severity.

The reviewer is correct. A time course study would have been very helpful. Unfortunately, there was only a limited number of double transgenic mice that was available for the study. This is why we had to choose a specific time point for treatment start. We chose 19 days after onset to reflect the time point of treatment start in MS patients that typically receive anti-CD20 therapy when their disease has already been ongoing for a while. To accommodate the reviewer’s comment we have added the following sentence to the discussion on page 15 of the revised manuscript: “In the current study, treatment was started in the stage of established disease. At that time, irreversible axonal damage has already occurred in EAE, and any treatment will not have an effect on disease severity unless it has neuroregenerative capacities. It remains to be determined whether differences between type I and II mAbs are more pronounced when administered shortly before or during acute disease, in which inflammatory processes prevail and neurodegeneration is only at its starting point”. 

2.2) Specific comments:

2.2.1) Introduction: 3 different subtypes of MS mentioned, however, there is a 4th type; progressive relapsing (PRMS) that is not mentioned. (see another comment in the discussion).

Following the reviewer’s comment we now also mention the progressive relapsing subset of MS on page 1 of the revised manuscript.

2.2.2) Results: Expressions used for demonstrating the results:

For example: Line 114, (beginning of the results) " no improvement…" , it actually describes that  there is no significant change in disease severity.

We have changed the sentence according to the reviewer’s suggestion.

2.2.3) Line 151 "no depletion…" when there are no significant changes/reductions in splenic B-cell content.

We have replaced the term “depletion” by “targeting”. Since there was indeed some degree of depletion we did not want to write “no depletion”.  

2.2.4) Statistics: p values are missing in the results. In most figures the statistic is difficult to follow; which group is compared with which.

We thank the reviewer’s comment. P-values are provided in the results section whenever statistical significance was reached. Please also check our figure legends, in which group comparisons are provided in detail along with the statistical test that was used for comparison. We now also clearly state when no statistically significant difference was observed.

2.2.5) Figure 3, demonstrates how the cell sorting experiments were performed, it is technical and does not provide real data, if at all, this figure may appear in the methods section.

Following the reviewer’s comment we have now moved Figure 3 to the methods section.

2.2.6) Discussion: The discussion is very short and not in depth, particularly it lacks explanations of why there is no correlation between the changes that occurred in cell population and myelin content and disease severity. Lately, the significant role of B-cells in MS and EAE received much attention and many studies have been performed and published investigated this subject, the authors did not elaborate enough on this topic in the discussion nor in the introduction.

We thank the reviewer for this comment. We have extended the discussion on pages and 14 and 15 in the revised manuscript. We have also moved the conclusion section to the end of the discussion, which now also provides an outlook.

Reviewer 3 Report

This is a study aimed mainly at comparing the effects of 2 types of anti-CD20 monoclonal antibodies in EAE, an animal model for multiple sclerosis. The main differences were found in the amount of B-cell infiltration into the brain which is of much interest in the treatment of SPMS and PPMS. Overall the manuscript is well written, well thought out and potentially important clincally. 

Minor suggestsion for discussion:

The use of mice with transgenic humanized CD20 is appropriate for the comparison of anti-human CD20 antibodies. As the authors state, it is difficult to compare the data with the mouse anti-CD20 since its effects probably involve binding to multiple ligands, including FC receptors mentioned in the Discussion. Are type I and type II mouse anti-CD20 antibodies available for further studies? 

One important difference between the EAE model presently used and clinical MS is the time course. In the relatively short time frame examined it is not reasonable to find reduction in antibody levels nor a clinical effect. MOG EAE models produce relatively long term effects over months in some studies.  Longer observation periods could potentially produce cleared histological effects and may be looked into in future studies. The authors should comment on the time frame.  

Author Response

3.1) The use of mice with transgenic humanized CD20 is appropriate for the comparison of anti-human CD20 antibodies. As the authors state, it is difficult to compare the data with the mouse anti-CD20 since its effects probably involve binding to multiple ligands, including FC receptors mentioned in the Discussion. Are type I and type II mouse anti-CD20 antibodies available for further studies? 

We thank the reviewer for this question. Unfortunately, type I and type II mouse anti-CD20 antibodies are not available for further studies. This is why we needed to use the double transgenic mouse line.

3.2) One important difference between the EAE model presently used and clinical MS is the time course. In the relatively short time frame examined it is not reasonable to find reduction in antibody levels nor a clinical effect. MOG EAE models produce relatively long term effects over months in some studies.  Longer observation periods could potentially produce cleared histological effects and may be looked into in future studies. The authors should comment on the time frame.  

We have included this helpful suggestion in our conclusion section on page 15 of the revised manuscript.

Round 2

Reviewer 2 Report

See attached file 
